# Hyperspectral imaging as an objective diagnostic tool for secondary lymphedema in breast cancer patients

Martin Weiss [1,2] ✉, Ovidiu Jurjuț [1], Astrid Ehrhardt[3], Anaclara Herholz[4], Anna Seller[1], Adrien Daigeler[4],
Markus Hahn[1], Mario Marx[3], Lukas Schimunek[1] & Wiebke Eisler[4]

## Abstract

**Background** Secondary lymphedema is a common complication after breast cancer treatment. Traditional diagnostic techniques often lack objectivity, are prone to operator bias, and show limited reproducibility. There is a need for diagnostic methods that are reliable, noninvasive, and suitable for monitoring disease progression. This study (German Clinical Trials Register, DRKS-ID: DRKS00027366) aimed to evaluate hyperspectral imaging as a tool for detecting and classifying secondary lymphedema.

**Methods** We conducted a prospective multicenter study including 58 women with unilateral secondary lymphedema. Lymphedema severity was categorized according to the International Society of Lymphology. Each participant underwent conventional assessments, including limb circumference measurements, two-point discrimination tests, joint mobility evaluation, and ultrasound imaging of skin and subcutaneous tissue, plus gave patient-reported outcomes. Hyperspectral images were collected using the TIVITA Tissue system, which captures light reflectance from 500 to 1000 nanometers. From these data, the lipid-to-water ratio and the TIVITA tissue water index were calculated to evaluate tissue changes related to lymphedema.

**Results** Here, we show that conventional methods correlate moderately with lymphedema stage, with ultrasound measurements of skin and subcutaneous thickness showing the strongest associations. Hyperspectral imaging shows strong correlations, with interlimb differences in lipid-to-water ratio and tissue water index outperforming conventional methods. These measurements highlight clear fluid accumulation in the affected limbs.

**Conclusions** Hyperspectral imaging provides reproducible, objective, and noninvasive assessment of secondary lymphedema. These findings support its potential as a diagnostic and monitoring tool to improve staging and treatment evaluation.

## Plain language summary

Secondary lymphedema is swelling of a limb and is a frequent side effect of breast cancer treatment. Current diagnostic methods can be subjective and inconsistent. We assessed whether hyperspectral imaging, which measures light absorption in the skin to estimate tissue water and fat levels, could be a more reliable alternative. Fifty-eight women with one arm affected by lymphedema were examined using standard tests, including measuring the arm circumference, skin sensibility, joint mobility, ultrasound imaging, and patient-reported outcomes, as well as hyperspectral imaging. Hyperspectral imaging showed stronger correlations with disease stage than conventional methods. Our results suggest hyperspectral imaging may offer an objective, noninvasive, and more reliable way to diagnose and monitor secondary lymphedema. The study was carried out at the Department of Women's Health in Tübingen, at the Department of Hand, Plastic, Reconstructive and Burn Surgery, BG Unfallklinik Tübingen and in the Clinic for Plastic, Reconstructive and Breast Surgery at the Elblandklinikum Radebeul.

Lymphedema of the upper extremity is a chronic inflammatory disease of the dermal interstitium caused by primary (hereditary) or secondary (acquired) damage to the lymphatic drainage system, including the initial lymphatic vessels, precollectors, lymph collectors, lymph trunks, and/or lymph nodes. Consequently, an insufficient lymphatic drainage leads to accumulation of proteins, fat, collagens, and other factors, causing an inflammatory fibrotic reaction with infiltration of monocytes, fibroblasts and adipocytes into the skin and subcutis. Contrary to primary LE, which is mostly due to a genetic predisposition, secondary LE can result from acquired diseases or injuries, such as operative therapies. For breast cancer, refined surgical techniques of lymph node removal have reduced the incidence of secondary LE to 19.9% after axillary dissection and 5.6% after sentinel node biopsy[1]. Moreover, obesity has an inducing and aggravating influence on the development of secondary LE[2]. Depending on the stage, LE can manifest itself

[1]Department of Women's Health, Eberhard Karls University of Tübingen, Tübingen, Germany. [2]NMI Scientific Medical Institute at the Eberhard Karls University of Tübingen, Reutlingen, Germany. [3]Clinic for Plastic, Reconstructive and Breast Surgery, Elblandklinikum Radebeul, Radebeul, Germany. [4]Department of Hand, Plastic, Reconstructive and Burn Surgery, BG Unfallklinik Tübingen, Eberhard Karls University of Tübingen, Tübingen, Germany.
✉e-mail: martin.weiss@med.uni-tuebingen.de

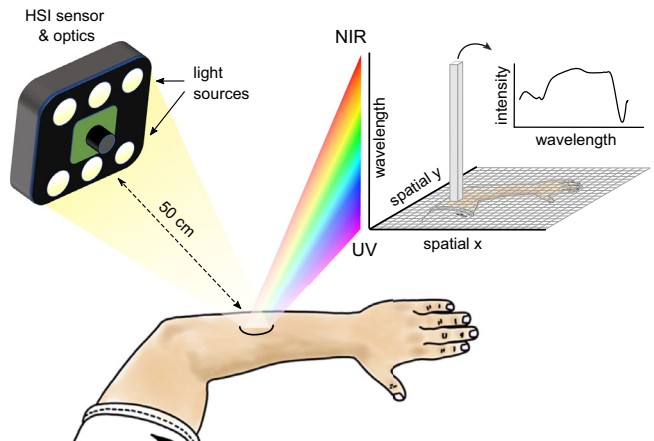

**Fig. 1 |** Schematic representation of the hyperspectral imaging setup and acquired data format.

symptomatically and asymptomatically. Typical symptoms may include pain, tension, and heaviness, as well as swelling and functional limitations. The staging according to the International Society of Lymphology[3] is currently based on anamnesis (also to clarify the etiology) and clinical examination with circumference measurements, survey of the range of motion and sensitivity, as well as inspection and palpation for localization. Furthermore, recording the "patient-reported outcome" evaluates associated long-term symptoms and reduced quality of life[4,5]. The valid assessment of the degree of illness is of great importance both for the selection of a suitable therapy and for the adequate provision of material and monetary assistance by the general public. However, the above-mentioned procedures are very susceptible to misjudgments and lack the necessary objectivity. Objective diagnostics that are currently only performed outside of routine and for special questions include sonography, functional lymphoscintigraphy, indirect and direct lymphangiography, fluorescence microlymphography, indocyanine green lymphography ("off-label use"), computed tomography, and magnetic resonance imaging. Indirect detection methods include the water displacement technique, 3D perometry, and bioimpedance spectroscopy. All of these require specialists for measurement and data interpretation, are mostly time-consuming, poorly evaluated, and cost-intensive.

In recent years, there have been increasing efforts to apply hyperspectral imaging in different areas of healthcare, including tissue differentiation, blood perfusion, wound vascularization, and inflammation[6–10]. Unlike a regular image, an HSI image contains a spectrum with wavelength-specific light reflectance intensity at each $(x, y)$ position. This results in a three-dimensional $(x, y, \lambda)$ spectral data cube behind every two-dimensional digital image (Fig. 1), where $x$ and $y$ correspond to spatial position and $\lambda$ corresponds to wavelength. Since different tissue types affect light reflectance at specific wavelengths, this technique enables analysis of tissue properties beyond the visible spectrum. HSI spectrometers are, in general, small-sized, transportable and comparatively inexpensive systems. To date, there is a limited number of medically approved HSI systems available, one of which is the TIVITA® Tissue device (Diaspective Vision GmbH, Pepelow, Germany).

The objective of this controlled, prospective, multi-center proof-of-principle study is to assess the suitability of HSI for contactless and objectifiable detection and classification of secondary LE using the TIVITA® Tissue device. The results show significant spectral differences between lymphedematous and healthy tissue, establish reliable classification parameters, and confirm HSI's potential as a valuable diagnostic tool for routine clinical assessment of upper extremity lymphedema.

## Methods
### Study design and patient enrollment
Female patients with unilateral secondary LE after breast cancer were enrolled at three study centers: (i) Department of Women's Health

Tübingen at the Eberhard Karls University Tübingen, (ii) Department of Hand, Plastic, Reconstructive and Burn Surgery, BG Unfallklinik Tübingenand (iii) Clinic for Plastic, Reconstructive and Breast Surgery, Elblandklinikum Radebeul. Patients of female gender, aged between 18 and 80 years, were included in the study. All participants received detailed information about the procedures and provided written informed consent. Exclusion criteria included thrombosis, bilateral LE, and lack of written informed consent.

### Ultrasound measurements
Ultrasound measurements were performed using standard ultrasound systems, depending on the location. In Tübingen, a Sonosite PX Ultrasound system (FUJIFILM Sonosite, Bothell, WA, USA) equipped with the L19-5 linear transducer was utilized. In Radebeul, three systems were employed: the Philips Affiniti 70 Ultrasound system (Philips Ultrasound Inc., Bothell, WA, USA) with the eL18-4 linear transducer, the GE Voluson S8 Ultrasound system (GE Ultrasound Korea Ltd., Seongnam, South Korea) with the 12L-RS linear transducer, and the GE Voluson E6 Ultrasound system (GE Healthcare Austria GmbH & Co OG, Tiefenbach, Austria) system with the 11L-D linear transducer. Ultrasound imaging examinations were performed by trained researchers to compare standardized sites on both arms: (i) the metacarpal region, (ii) the forearm, 10 cm distal to the lateral epicondylus, and (iii) the upper arm, 15 cm proximal to the lateral epicondylus. The participant was laid supine with the respective arm extended to improve the quality of the image and visualization. Ultrasound images were recorded to measure the cutis and subcutis thickness. Data are expressed as a percentage change (value$_{affected}$ / value$_{unaffected}$ × 100).

### Measurement of circumference, sensibility, and mobility
The location of assessment sites was standardized to the identical arm measuring sites described above. The arm circumference was determined in cm. The difference between the affected and unaffected sides was expressed as a percentage change (value$_{affected}$/value$_{unaffected}$ × 100) to exclude effects of physiological intra- and interindividual variety. For the TPD and arm mobility assessments, a direct comparison of absolute results was possible, and the change between sides was quantified by the difference between the affected and unaffected side measurements (value$_{affected}$ − value$_{unaffected}$).

### Patient-reported outcomes
Secondary LE were assessed according to the Lymph-ICF-UL and EORTC QLQ-BR23 questionnaires, which incorporate five multi-item scales to assess problems in functioning (i.e., impairments in function and activity limitations and participation restrictions) and symptoms (body image, sexual functioning, systemic therapy side effects, breast symptoms, and arm symptoms). High scores for the functional scales represent a high/healthy level of functioning, whilst high scores for the symptom scales represent a high level of symptomatology or problems. These questionnaires are descriptive tools[6,11]. Questions were rated on a scale from 1 to 4 and grouped into categories. To obtain a score for each category, the average score of the questions within a category was calculated and scaled from 0 to 100. Questions that were not answered were excluded from the category score. Since the questionnaire results were patient-based and not related to a specific body side, the category scores were used directly to assess changes associated with LE stage.

### Acquisition of hyperspectral images
Hyperspectral images were acquired using the TIVITA® Tissue System (Diaspective Vision, Germany), a medically approved HSI system with a cost below $10.000. Measurements were carried out in a darkened room to minimize external light interference. The system was calibrated with a color checking palette at the beginning of each measuring day. Patients were instructed to rest their arm on a black surface, and the camera of the imaging system was positioned at a fixed distance of 50 cm from the imaging area. Imaging was performed at three specific positions on the arm: metacarpal backhand, 10 cm below the epicondylus lateralis, and 15 cm above the

epicondylus lateralis. These positions corresponded to locations where arm circumference and cutis/subcutis thickness had been previously measured. Measurements were taken at all three positions, regardless of whether the site was diagnosed as affected by LE. To ensure accuracy and redundancy, each measurement was repeated three times, allowing for the selection of the best-framed image for subsequent analysis.

Each HSI measurement yielded a set of images that represent the reflectance of light at given wavelengths. The resulting 3-dimensional matrix of values is also referred to as a hypercube. The spectral imaging resolution of the TIVITA® Tissue System is $640 \times 480$ pixels, and the wavelength range is from 500 to 1000 nm in steps of 5 nm, yielding thus hypercubes of size $640 \times 480 \times 100$.

## Analysis of hyperspectral images

To visualize different tissue properties, the TIVITA® Tissue System provides a set of indices that measure the presence of water (TWI–tissue water index), oxygen ($StO_2$–tissue oxygenation), hemoglobin (THI–tissue hemoglobin index), lipids (TLI–tissue lipid index) and the perfusion of tissue (NIR–near infrared perfusion). In the current study, TWI and TLI were used to test the correlation with LE stage. $StO_2$ was also used together with TWI to distinguish body parts from the surrounding background in HSI images. The indices were computed from the spectral signal within individual wavelength bands and scaled to yield values between 0 and 100. The exact formulas and scaling factors are proprietary to the manufacturer of the HSI system however, many details about the algorithms can be found in Holmer et al.[7]. In summary, TWI is a scaled ratio between the average signal in the 875–895 and 950–975 nm bands, TLI is a scaled mean of the second derivative of the signal in the 925–935 nm band and $StO_2$ is a scaled ratio between the second derivative of the signal in the 575–590 and 740–780 nm bands.

The LWR is defined as the ratio of the average signal in the 925–935 and 965–985 nm wavelength bands, similar to TLI and TWI. These bands correspond to the light absorption intervals of lipids[7,8] and water[7,9], respectively.

For the background separation algorithm, a combination of TWI and $StO_2$ was selected, as these two tissue properties, water content and oxygenation level, proved most effective at separating imaged body parts from the surrounding background. First, the TWI and $StO_2$ indices were computed for each pixel in the HSI image to produce corresponding TWI and $StO_2$ images. These were then filtered using a Gaussian kernel of size (5, 5) pixels to reduce noise. The index images were then normalized to their maximum value and summed up. The normalized summed values range from 0 to 2, with 0 representing low TWI and $StO_2$ indices for that pixel and 2 representing high indices. For the vast majority of measurements, the histogram of the normalized summed values was bimodal, with one peak corresponding to background pixels and the other to the body part. This enabled Otsu's adaptive binarization method to be applied to determine the optimal threshold for separating the two peaks. In instances where the histogram was not bimodal, a fixed threshold of 1 was used. This approach also worked well for processing measurements where no body part was present in the image, or where the body part occupied most of the image. The resulting threshold was then used to generate a mask of the body part for each measurement. Based on the extent to which the body part covered the central area of the image, one measurement out of the three repeats was selected using the body mask.

The hyperspectral data were evaluated at the center of the image, within a circular area with a diameter of 100 pixels, corresponding to 4.6 cm. Spectra corresponding to body pixels in this area were normalized to their summed intensity using L1 normalization and averaged to obtain a representative spectrum for each measurement. Except the standard calibration with the color checking palette, performed at the beginning of each measurement day, no additional calibration measurements were considered for the normalization of spectra. TWI, TLI, and LWR values were computed from the average spectrum. To account for potential variations in global lighting between measurements, indices were calculated separately for each

side of the body. The difference between the affected and unaffected sides was calculated by subtracting their respective index values.

For a grading of measurements based on LWR, hierarchical clustering with a cluster size of $k = 3$ was applied, resulting in three groups: low, mid, and high HSI stages. When $k = 2$ was used, the low and mid groups merged, while the high group remained unchanged. Using $k = 4$ further subdivided the high group. A cluster size of $k = 3$ was chosen as it best captured the visual grouping of the values.

## Statistics and reproducibility

To test for statistical significance between the distributions of ΔLWR values per LE stage, the distributions were first tested for normality using the normaltest function from the SciPy library (version 1.11.3). The values were not normally distributed for stage 2 LE. For stages 0, 1, and 3, the small sample size ($n = 10$, 9, and 14, respectively) prevented a confident assessment of the underlying distribution shape. Due to these constraints, the two-sided nonparametric Mann–Whitney U test was used to test for significant differences in ΔLWR values between adjacent LE stages using the mannwhitneyu function from the same library. Spearman's correlation coefficient (function corr from the Pandas version 2.1.1 library) was used to assess the relationship between LE stage and the metrics used to describe the measurements.

The patient cohort was sufficiently large to ensure adequate power for detecting moderate correlations between the different measurements and clinical disease stage. Power calculations based on the Fisher z transform indicated that with $n = 58$, $\alpha = 0.05$, and 80% power, the study is able to detect correlations of $r \geq 0.36$.

## Results

The study was carried out between December 2021 and June 2023. Fifty-eight female patients with unilateral secondary LE after breast cancer treatment were measured at (i) the Department of Women's Health Tübingen at the Eberhard Karls University Tübingen, (ii) the Department of Hand, Plastic, Reconstructive and Burn Surgery, BG Unfallklinik Tübingen and (iii) the Clinic for Plastic, Reconstructive and Breast Surgery, Elblandklinikum Radebeul. Patients ranged from 31 to 80 years, with a mean age of 56 years. Their average BMI was 27.2 kg/m² (range: 18.5–43.4 kg/m²). Based on the International Society of Lymphology's criteria[10] 11 patients were classified as stage 0 LE, 13 as stage I, 28 as stage II and 6 as stage III (Fig. 2). The stages correspond to: 0-no visible edema or pitting, I-visible edema with reversible pitting, II-edema with pitting, III-elephantiasis with skin lesions and relapsing infections. Figure 2 shows examples of hands for patients for each LE stage, along with the distribution of patients across the study centers. Patients with multiple stages were categorized based on the highest stage present.

## Values of circumference, sensibility, mobility, quality of life assessment and skin thickness partially correlate to secondary LE stage

First, secondary LE characterization was conducted by comparison of the affected and unaffected extremities using conventional methods, including circumference, TPD test, and joint mobility. To obtain comparable values for changes in circumference across different body regions, differences between the affected and unaffected sides were expressed as a relative change with respect to the unaffected side and reported as percentages. Figure 3a shows the percentage change in circumference measured metacarpal, 10 cm below and 15 cm above the epicondylus lateralis, the elbow joint, but only for locations and their contralateral side labeled as affected by physicians. TPD sensibility was measured at the tips of the index and little fingers and was reported as the difference in discrimination threshold between the affected and unaffected sides. Figure 3b shows the distribution of sensibility across LE stages for both index and little fingers. Joint mobility was measured at the shoulder, elbow, wrist, and thumb, and reported as the difference in maximal motion angle between the affected and unaffected sides, ranging from 0° to 180°. Assessed movements included the shoulder

## a Lymphedema examples

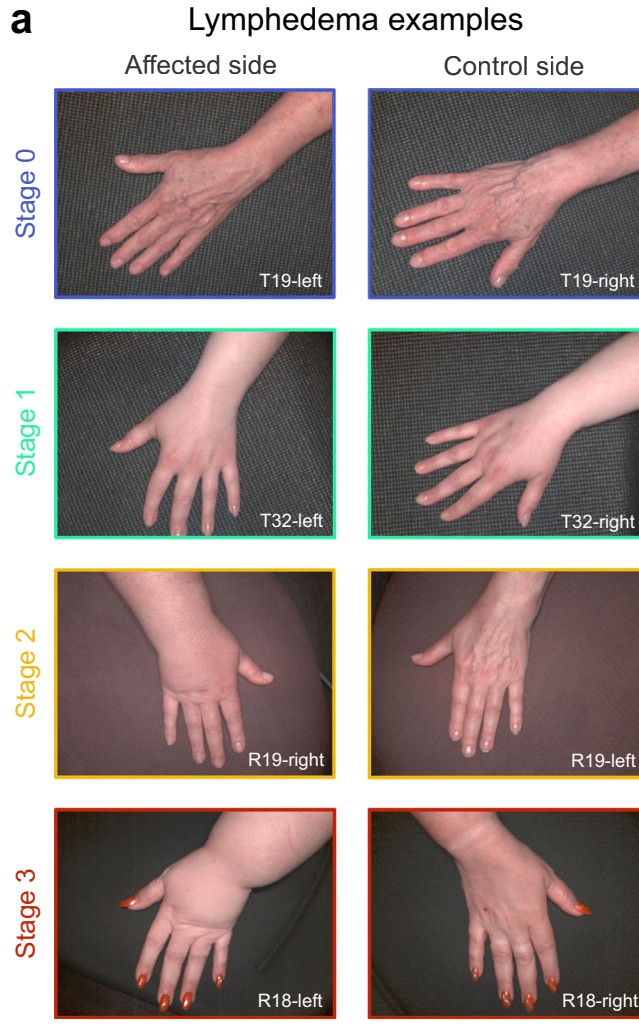

## b Patients distribution

| Data center | Lymphedema patients | | | |
|---|---|---|---|---|
| | **Stage 0** | **Stage 1** | **Stage 2** | **Stage 3** |
| Tübingen | 11 | 10 | 6 | 2 |
| Radebeul | 0 | 3 | 22 | 4 |
| Total | **11** | **13** | **28** | **6** |

**Fig. 2 | Lymphedema stages and patient distribution. a** Examples of patient's hands affected by LE (left column) compared to the unaffected contralateral side (right column) at different stages of the disease. **b** Summary of patients from the current study, categorized by LE stage and the center where data were collected.

(glenohumeral joint): abduction/adduction, retroversion/anteversion, external/internal rotation at close-fitting and 90° abducted upper arm; the elbow (humeroulnoradial joint): extension/flexion, supination/pronation; the wrist (radiocarpal joint) extension/flexion, radial/ulnar adduction; and the thumb (metacarpophalangeal joint): extension/flexion. Independent of the LE stage, the thumb pad could reach all fingertips on both sides, and fist clenching and finger stretching were unrestricted on both sides. Figure 3c shows the differences in movement angle for a range of movements that correlated most strongly with the stage of LE. Results for the full set of movements are available as supplementary material (Fig. S1).

Next, the correlation of the LE stage to the patients' quality of life was assessed using the LYMPH-ICF-UL and EORTC QLQ-BR23 questionnaires. The first focuses on evaluating issues like pain, mobility, and

daily functioning, while the latter is designed specifically for breast cancer patients and focuses on aspects such as body image, treatment side effects, and overall well-being. Figure 3d shows the questionnaire scores most correlated with LE stage, the strongest being related to arm symptoms (Spearman's $r = 0.54$). Results for the full set of questionnaire scores are available as supplementary material (Fig. S2).

Last, subcutaneous swelling as a result of the accumulation of protein-rich fluid in tissues was evaluated using ultrasound. The thicknesses of the cutis and subcutis layers were recorded at the same positions where the arm circumference was measured, using standardized ultrasound examination and compared between affected and unaffected extremities. Like in the case of circumference measurements, only locations and their contralateral side labeled as affected by physicians were considered. The change in cutis and subcutis thickness was expressed as a percentage, relative to the thickness of the unaffected side. Figure 3e shows increased cutis and subcutis thickness for the affected side with higher LE stages.

To summarize, Fig. 3f shows the correlation of all the measurements described above with the diagnosed LE stage. While some measurements are correlated to LE stage, their correlation is at best "moderate", with the highest values for the percentual change in cutis and subcutis thickness (Spearman's $r = 0.57$ for cutis and $r = 0.56$ for subcutis, respectively).

### Hyperspectral imaging reveals stronger correlation to lymphedema stage

To assess whether HSI information provides a better indicator for LE stage the same arm positions shown in Fig. 3a were measured with a TIVITA® Tissue System (Fig. 4a) on both affected and unaffected sides. The system records hypercubes of size depicted in Fig. 4b and can display the distribution of tissue properties as defined by the TIVITA indices (Fig. 4c).

Before analyzing the raw spectra, an algorithm was applied to the HSI images to separate body areas from the background. This ensured that non-tissue spectra were excluded from further analysis. First, the normalized TWI and $StO_2$ indices were summed, and a histogram of these values was constructed. Then, Otsu's thresholding method was applied to determine a cutoff threshold in the histogram. This was possible as the majority of histograms exhibited a bimodal distribution, which indicated good separation between body and background. The cutoff threshold was then used to generate an image mask corresponding to the body part. Finally, the mask was applied to the associated hypercube, isolating spectra that originate from the body region from those of the surrounding background (Fig. 4d).

The raw spectra were normalized to their intensity using L1 normalization and averaged over a patch of tissue, approximately 4.6 cm in diameter, located in the center of the image for both the affected and unaffected side (Fig. 4e). These locations corresponded to those used for the arm circumference and the cutis/subcutis thickness measurements. Figure 4f shows the average spectra for the center patches of the examples in Fig. 4e. In this particular case, the average spectra of the affected and unaffected sides looked quite similar across a broad range of wavelengths, with notable differences around 920 and 970 nm. These wavelength bands correspond to the absorption bands for lipids[7,8] and water[7,9], respectively. The differences became more pronounced when the spectrum of the affected side was divided by that of the unaffected side, resulting in a ratio that showed a peak around 920 nm and a dip around 970 nm. In physical terms, this translates to a reduction in lipid content accompanied by an increase in water content on the affected side.

To quantify this effect across patients and relate it to the diagnosed LE stage, the LWR was introduced. The LWR is defined as the ratio of the signal in the lipid absorption band (920–935 nm) to the signal in the water absorption band (965–985 nm), as shown in Fig. 4g. The difference in LWR between the affected and unaffected sides (ΔLWR) quantifies the strength of the previously described effect. Low values corresponded to minimal differences in the lipids and water spectral bands, while high values corresponded to pronounced differences.

The ΔLWR was calculated for all HSI measurements of arm locations that were diagnosed as affected by a physician. The resulting values are

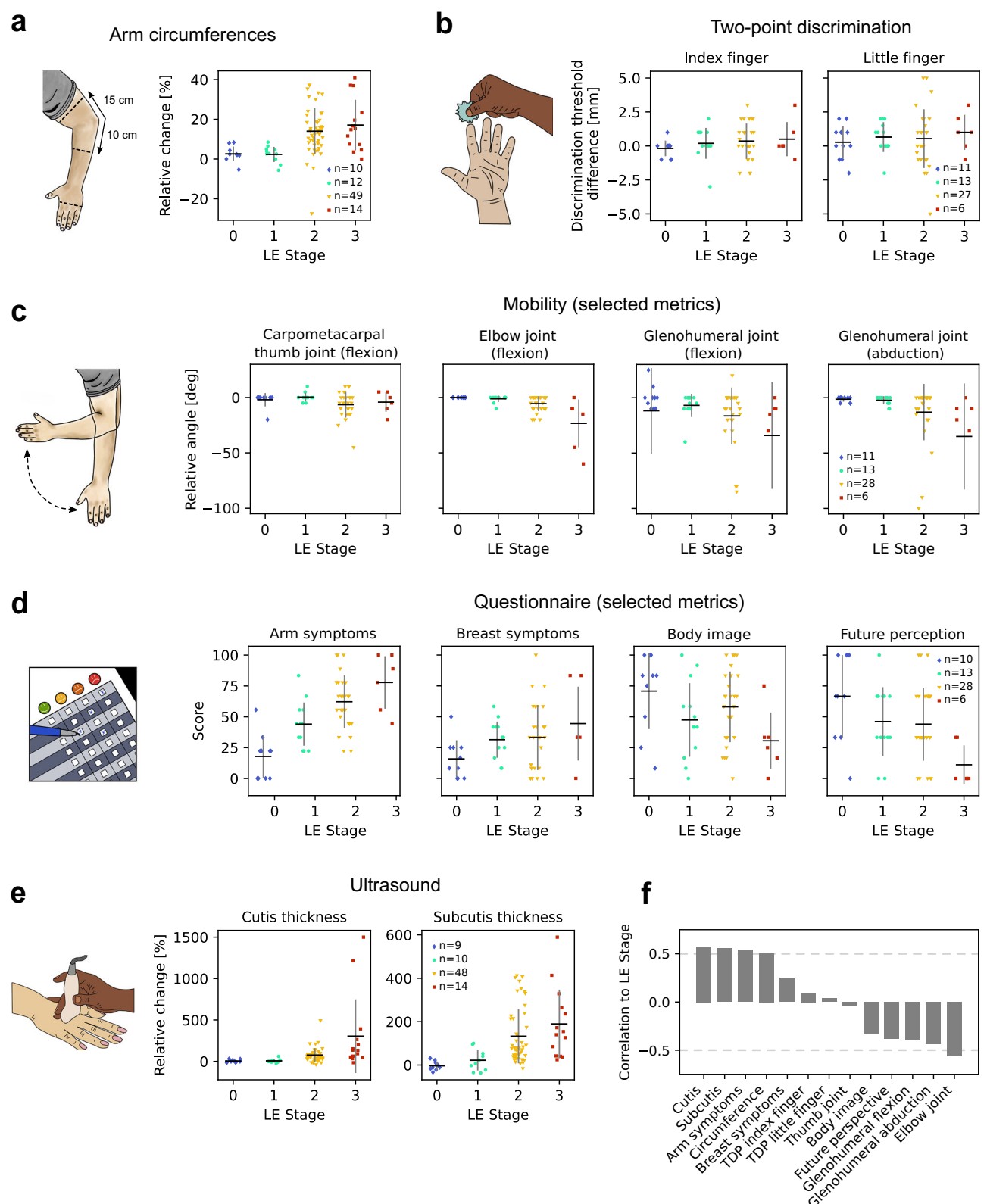

shown as a histogram in Fig. 4h. To obtain a grading of HSI measurements, similar to the LE grading but based on the LWR, hierarchical clustering with $k = 3$ was applied, and the resulting groups were termed low, mid, and high HSI stages, corresponding to low, mid and high ΔLWR values. Figure 4i left panel shows the ΔLWR values of HSI measurements taken at different LE stages, with colors corresponding to the HSI stages defined in Fig. 4h.

Measurements from patients with LE stages 0 and 1 consistently displayed low HSI stages. In contrast, measurements from patients with stage 2 LE exhibited the whole range of HSI stages, indicating heterogeneous spectral effects within this group. The final group, consisting of stage 3 LE patients, displayed only mid to high HSI stages. The same ΔLWR values can also be displayed by HSI stage, with colors indicating the diagnosed LE stage, as

**Fig. 3 | Patient measurements by LE stage. a** Percentage increase in circumference between the affected and unaffected sides, measured at the middle of the hand, 10 cm below the elbow joint, and 15 cm above the elbow joint. **b** Difference in the TPD threshold between the affected and unaffected sides, measured at the index and little fingers. **c** Difference in movement angle between the affected and unaffected sides, measured at the fingers, arm and shoulder. Only a subset of movements is shown here, see supplementary materials for the full range of movements. **d** Quality of life scores reported by patients, quantified using the LYMPH-ICF-UL and EORTC QLQ-BR23 questionnaires. Only a subset of metrics is shown here, see

supplementary materials for the full set of scores. **e** Percentage increase in cutis and subcutis thickness between the affected and unaffected sides, measured at the same locations as in (**a**). **f** Spearman's correlation (*r*) between LE stage and the measurements from (**a–e**). Note: Data points in (**a, e**) include only measurements from arm locations, and their contralateral side, confirmed as affected by physicians. Data points in **b–d** represent measurements in individual patients. In **a–e**, horizontal lines indicate the means, while vertical bars represent the standard deviation of the distributions. Sample sizes are shown in the last graph of each panel.

shown in the right panel of Fig. 4i. This visualization method highlights that mid and high HSI stages included only measurements from patients with stage 2 and 3 LE. In contrast, the low HSI stage included measurements from patients with stage 0, 1, and 2 LE. This suggests that some patients diagnosed with stage 2 LE did not exhibit physiological tissue changes measurable with the HSI method used in this study.

To place HSI in the context of objective approaches for diagnosing LE stage, the correlation between HSI-derived metrics and the diagnosed LE stage was calculated. Figure 4j shows the Spearman's *r* correlation between LE stage and HSI stage, $\Delta$TWI, $\Delta$TLI, as well as the two most correlated metrics from Fig. 3, cutis thickness and elbow joint mobility. The $\Delta$TWI and $\Delta$TLI were calculated the same way as the $\Delta$LWR, but using the TIVITA-defined TWI and TLI values instead of LWR. HSI stage and $\Delta$TWI showed the highest degree of correlation with LE stage (*r* = 0.64 and *r* = 0.63 respectively), surpassing all metrics shown in Fig. 3 where the maximal correlation was *r* = 0.57. The similar correlation values of these metrics, along with the HSI stage's reliance on both water and lipids, suggest that the increase tissue water content was the primary factor driving the correlation with the diagnosed LE stage. This conclusion was further supported by the lower correlation with the $\Delta$TLI, indicating that the decrease in lipid tissue alone could not explain LE grading as well as the water content did.

In summary, both the HSI stage metric and the $\Delta$TWI are objective metrics that show the strongest correlation with LE stage among all other metrics tested within the scope of this study.

### Ethics and inclusion

This multicenter-center, one-arm, prospective observational study was designed in accordance with the Declaration of Helsinki, approved by the local ethics committee at the Medical Faculty of the University of Tübingen (044/2021BO2) and the ethics committee of the Landesärztekammer Sachsen (EK-BR-126/21-1) and registered at the German Clinical Trials Register (DRKS.de, DRKS-ID: DRKS00027366, date of registration: 08.12.2021). Staff recruitment for the project followed the regulations of the respective institutions and complied with the German Equal Treatment Act. All authors are affiliated with and local to the centers where the data were collected.

### Discussion

This study evaluates the potential of HSI as a diagnostic tool for secondary LE following breast cancer treatment. Traditional diagnostic techniques for LE, including circumference measurement, ultrasound imaging, and patient-reported outcomes, are often criticized for their lack of objectivity, susceptibility to operator bias, and limited reproducibility. Moreover, these methods frequently fail to capture the complex pathophysiological changes underlying LE, such as alterations in tissue composition and fluid distribution. Ultrasound imaging has been studied but has not yet been incorporated into standard care, most likely due to lack of practicability, investigator dependency and high logistical effort. Furthermore, patient-reported outcomes are subjective, highly dependent on daily condition and may not accurately reflect disease severity. These limitations collectively result in diagnostic uncertainty, delayed interventions, and, in some cases, mismanagement in the form of under- or overtreatment of the condition.

HSI enables the contactless, noninvasive, and radiation-free characterization of tissue samples within a matter of seconds. This technology provides a high-resolution, two-dimensional representation of a surface,

incorporating information from underlying layers. Using the clinically approved TIVITA® Tissue system (500–995 nm), HSI has shown remarkable versatility, from monitoring peripheral arterial disease to optimizing surgical outcomes[12–15]. It has been successfully applied in visceral and reconstructive surgeries for intraoperative blood flow visualization, enhancing postoperative results, and guiding timely interventions in cases of compromised perfusion[16–21]. Beyond surgical applications, HSI has proven effective in burn depth assessment[22,23], wound monitoring[24], and non-invasive differentiation of arterial and venous pathologies through TWI analysis[15,25]. Initially designed for wound and perfusion diagnostics, the TIVITA® Tissue System has demonstrated great potential in the field of tissue differentiation, as indicated by a recent study[26]. In this study, Studier-Fischer et al. collected a HSI dataset from the visceral organs of 46 pigs and obtained 9059 HSI images in total. Based on this, the authors developed a comprehensive tissue atlas with spectral fingerprints of 20 distinct porcine organs and tissue types. Utilizing mixed model analysis and deep neural networks, the authors further demonstrated the feasibility of fully automated tissue differentiation across the 20 organ classes with an accuracy exceeding 95%. Furthermore, our working group evaluated the ability of the TIVITA® Tissue System to differentiate high-grade cervical intraepithelial neoplasia from normal cervical tissue, without contact and marker-free, within a prospective clinical proof-of-principle study[27].

In this study, we extended the scope of HSI by introducing an additional index, the lipid-water ratio, alongside TWI and TLI, to refine secondary LE diagnostics. By integrating this index with hierarchical clustering, we proposed a HSI-based secondary LE classification, opening a promising avenue for precise and objective disease staging compared to conventional techniques. The $\Delta$LWR metric, which quantifies differences in lipid and water spectral bands between affected and unaffected extremities, showed a notable progression across LE stages. Patients with stage 0 and stage 1 LE consistently presented low $\Delta$LWR values, suggesting that this metric could not detect early-stage LE. In contrast, patients with stage 2 and stage 3 LE exhibited a broader range of values, reflecting increased tissue heterogeneity with advancing disease and the potential existence of pathological subcategories in the later stages of the disease. Compared to circumference or cutis/subcutis thickness measurements, which showed moderate correlations with disease stage (*r* = 0.57 and *r* = 0.56, respectively), $\Delta$LWR values and the derived HSI-based staging provided a more robust and consistent representation of LE severity. While early-stage LE is dominated by interstitial fluid, later stages involve adipose and fibrotic changes. Notably, adipose tissue in chronic LE may also retain water due to inflammation-induced permeability[28], potentially explaining the predominantly dispersed $\Delta$LWR values in stage 2. This underscores the advantage of HSI in capturing tissue-specific pathological changes—such as localized water retention and lipid redistribution—that remain undetectable by traditional diagnostic tools.

The HSI approach proposed in this study addressed many limitations of traditional LE diagnostic tools. Besides the noninvasive nature of the method, using quantitative objective tissue biomarkers offers a high degree of measurement reproducibility, which is an advancement over traditional, less precise methods. This opens the possibility of identifying subclinical changes in LE stages, a feature that conventional methods lack. Moreover, an automated standardization of spectral analysis may further enhance the diagnostic consistency, reducing operator dependency, another key limitation in traditional approaches. Regarding hardware optimization, the results of this study suggest that only a few wavelengths are sufficient for

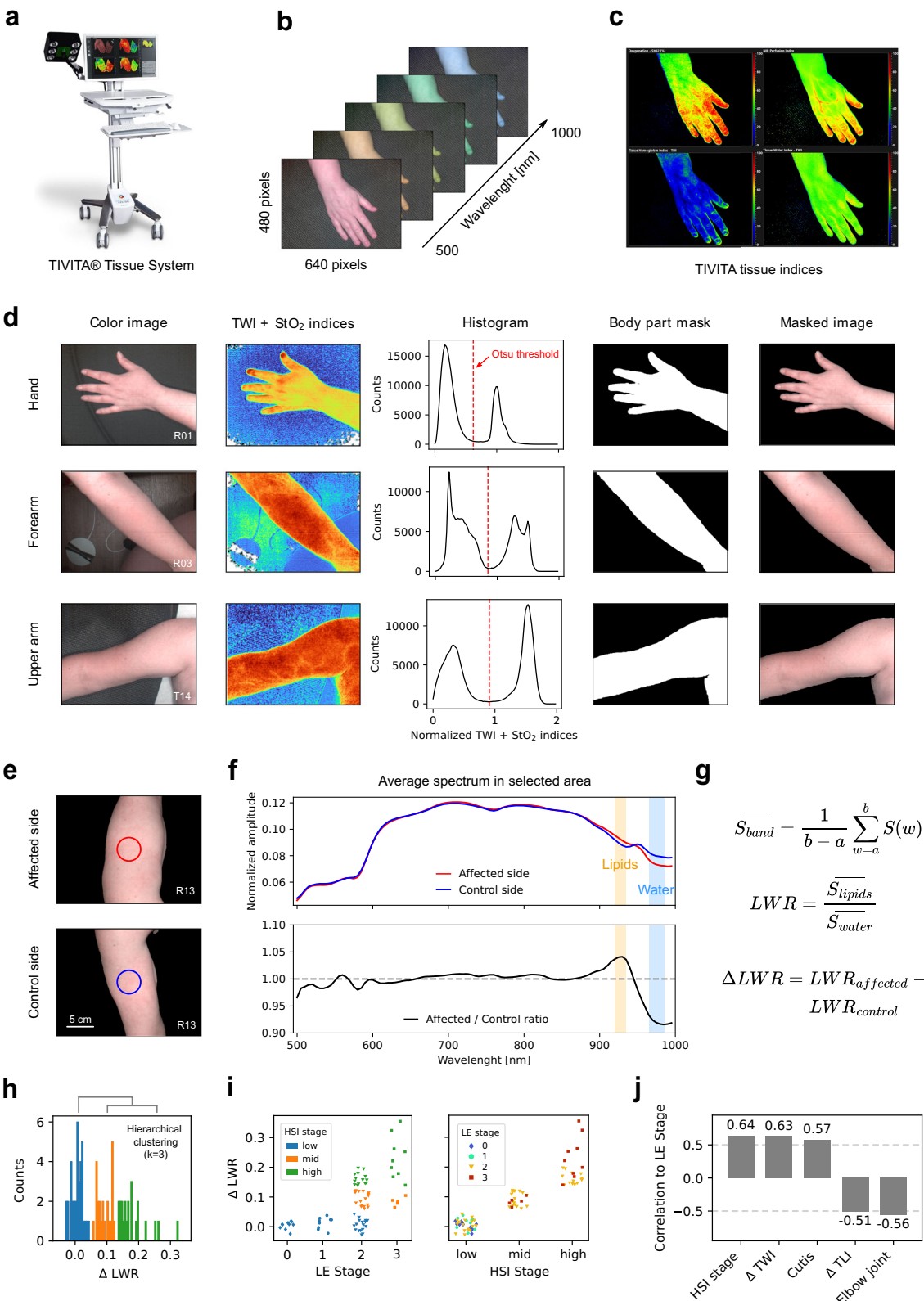

tissue characterization in this particular application. Therefore, one could envision developing more cost-effective systems that measure light only at the wavelengths of interest. This would provide clinicians in less resource-rich regions, who cannot afford a full HSI system, with access to the same enhanced diagnostic capabilities. Additionally, given the strong correlation between the proposed HSI-stage metric and disease severity observed in this

study, further developing a dedicated HSI-based classification system for secondary LE warrants serious consideration.

There are some weaknesses of the current study, such as the limited number of participants, which may not be sufficient to ensure robust generalizability of the findings, particularly for a condition with such diverse clinical manifestations as secondary LE. The study lacks longitudinal data to

**Fig. 4 | Hyperspectral imaging of lymphedema. a** TIVITA Tissue system used for hyperspectral imaging acquisition. **b** Parameters of one hyperspectral measurement. **c** Spatial distribution of TIVITA-defined indices describing tissue properties. From top-left to right: $StO_2$, NIR, THI, and TWI. **d** Steps of the algorithm for separating the patient's body from the background. From left to right: an image of the arm section, the sum of normalized TWI and $StO_2$ indices with blue/green hues corresponding to small values and orange/red hues corresponding to high values, the histogram of normalized summed indices showing a bimodal distribution that corresponds to background and body pixels, the mask of the arm section after applying Otsu thresholding to the histogram, and the masked image of the arm section. Rows represent individual examples for the hand, forearm, and upper arm. **e** Example of affected and unaffected sides of a patient's upper arm, with a marked circular area in the center indicating where hyperspectral data is averaged. **f** Upper panel: Average normalized hyper-spectra over the area indicated in (**e**) for the affected (red) and unaffected (dark blue) sides. Lower panel: ratio of the two spectra in the upper panel. The orange and light blue bands indicate wavelength bands sensitive to the presence of lipids and water, respectively. **g** Formulas for calculating the LWR. **h** Histogram of LWR differences between the affected and unaffected sides. Colors indicate value groupings into clusters after applying hierarchical clustering with $k = 3$ clusters. **i** ΔLWR between the affected and unaffected sides as a function of LE stage (left panel) and the hierarchical clustering grouping (right panel). Sample sizes were $n = 10, 9, 49$, and 14 for stages 0, 1, 2, and 3, respectively. "**" denotes $p < 0.01$ for the Mann–Whitney-U test (stage 0 vs. 1: $p = 0.17$; stage 1 vs. 2: $p = 0.008$; stage 2 vs. 3: $p = 0.001$), "n.s." stands for non-significant. Colors represent the clusters from (**h**) in the left panel and the LE stage in the right panel. **j** Spearman's correlation ($r$) between LE stage and hyperspectral imaging-derived measurements, as well as the best correlating values from Fig. 3f.

demonstrate how ΔLWR values correlate with the progression or regression of LE over time. The study did not consider the influence of patients' water intake behavior. Measurements were carried out before noon to minimize the influence of tissue swelling throughout the day, and patients with an indication to wear support stockings were asked to remove them the evening before and only put them on again after the measurement. However, this does not eliminate the potential impact of individual hydration levels on the results. Moreover, variables such as BMI, skin color, or comorbidities that might affect HSI measurements were not comprehensively addressed in the analysis at this stage. It is conceivable that some of the heterogeneity in ΔLWR values for stage 2 LE patients could be attributable to these variables. The proof-of-principle study does not intend to define clear threshold values for ΔLWR that could be used in clinical practice to differentiate between stages of LE. Future research should focus on validating this classification system across diverse patient populations and integrating additional spectral biomarkers to enhance diagnostic granularity. Artificial intelligence and neural networks may enhance HSI-based diagnostics by automating image analysis, identifying key tissue biomarkers, and predicting disease progression, thus improving diagnostic accuracy and speed. Additionally, machine learning models could optimize the selection of HSI parameters and support personalized treatment plans by learning from extensive, diverse patient data. To achieve this, large validated datasets, interdisciplinary collaboration and robust clinical trials are essential for successful implementation in clinical practice.

## Conclusion
This proof-of-principle study demonstrates that hyperspectral imaging using the TIVITA® Tissue system provides superior objective assessment of secondary lymphedema compared to conventional diagnostic methods. The hyperspectral-derived metrics, particularly the HSI stage and tissue water index differences, showed stronger correlations with lymphedema severity ($r = 0.64$ and $r = 0.63$) than traditional measurements, including circumference and ultrasound thickness assessments. The contactless, noninvasive nature of HSI technology, combined with its ability to detect tissue composition changes, offers significant advantages over current subjective and operator-dependent diagnostic approaches. These findings support the clinical potential of hyperspectral imaging as a reliable, reproducible tool for lymphedema detection, staging, and monitoring treatment response. Future research should focus on validating these results in larger, more diverse patient populations and developing standardized threshold values for clinical implementation.

## Data availability
The raw data that support the findings of this study are available from the corresponding author, M.W., upon reasonable request and subject to a non-disclosure agreement. Due to ethical and privacy considerations, the data cannot be made publicly available, as it includes images of body parts that may contain identifiable features, posing a potential risk to patient anonymity. Numerical data underlying the figures are included in the supplementary materials. Specifically, the source data for Figs. 3 and 4i, j are provided in Supplementary Data 1, and those for Supplementary

Figs. S1 and S2 in Supplementary Data 2. Both files are provided as Excel spreadsheets, with individual figure panels organized in separate sheets.

## Code availability
All analyses were performed using custom Python scripts, with support from widely used open-source libraries such as OpenCV (4.9.0.80), Scikit-learn (1.4.1.post1), SciPy (1.11.3), NumPy (1.26.1), and Pandas (2.1.1). The code is not publicly available, as it includes proprietary formulas for computing tissue indices, which are the intellectual property the manufacturer of the TIVITA® Tissue system. Sharing these formulas would require explicit permission from the manufacturer. However, the code can be made available upon request from MW, pending approval from the HSI system manufacturer and subject to a non-disclosure agreement.

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

## Acknowledgements
This work was supported by Diaspective Vision, Germany and Keller Medical, Germany (loaner of the TIVITA® Tissue System) with technical support.

## Author contributions
W.E. and M.W. conceived the study, designed and supervised the experiments. W.E. and A.S. screened the patient cohort, A.E. and A.H. collected the data, including patient measurements, ultrasound and hyperspectral imaging data. O.J. and M.W. analysed the data, prepared the figures and wrote the manuscript. A.D., M.H., M.M. and L.S. reviewed the manuscript.

## Funding

## Competing interests
The authors declare no competing interests.
