## [Transparent Peer Review file · Communications Medicine]

Hyperspectral imaging as an objective diagnostic tool for secondary lymphedema in breast cancer patients

Corresponding Author: Professor Martin Weiss

Version 0:

Reviewer comments:

Reviewer #1

(Remarks to the Author)

The authors investigated the potential of hyperspectral imaging (HSI) as a diagnostic tool for secondary lymphedema in breast cancer patients using the TIVITA HSI system. Their findings suggest that the lipid-to-water ratio (LWR) may serve as a promising objective marker for detecting lymphedema.

Here are some comments:

- 1) Acronyms such as HSI, TWI, TLI, and LWR are defined multiple times throughout the manuscript. It would improve clarity and flow to define each acronym only at its first occurrence and use the acronym consistently thereafter.
- 2) While the study utilized a commercial HSI system and analysis software, additional detail on how TWI and TLI are calculated would enhance transparency and reproducibility. A brief description of the algorithm or underlying principles would be beneficial.
- 3) The manuscript states that "raw spectra were normalized to their overall intensity." Please clarify the normalization process. Specifically, does it involve referencing to white and dark calibration measurements, or is another normalization method used?
- 4) If the LWR is derived from spectral information around 920 and 970 nm, could similar results be achieved using just two narrowband measurements at these wavelengths? If so, this raises the possibility of implementing a more cost-effective and portable device rather than requiring a full HSI system.
- 5) Given that hyperspectral image data were collected from 58 patients, it seems feasible to perform statistical analysis of the measured spectral data or derived index (LWR).
- 6) The descriptions of the hyperspectral imaging (HSI) procedures and results appear to be somewhat redundant between the Methods and Results sections. It is recommended to streamline these sections by removing repetitive content, ensuring that methodological details are clearly described in the Methods, while the Results section focuses on presenting key findings and interpretations.

Reviewer #2

(Remarks to the Author)

This manuscript reports a prospective, multi-center study, conducted by a very qualified group, to assess hyperspectral imaging (HSI) as a method for detection and classification of secondary LE.

The figures are very neatly prepared, and the writing is good.

The main finding of the study is that HSI images in 58 breast cancer-related study subjects showed Pearson's coefficients of 0.64 (compared to $r=0.57$ for skin thickness measured by ultrasound) correlating to LE stages.

Please check and perhaps replace some of the references, and do not rely on AI. For example, reference 2 is given to describe gynecological cancer-related LE, yet that paper reports on breast cancer-related LE, which is not considered gynecological. For reference 6, please use the ISL position consensus, not an anatomy book. Reference 7 does not discuss HSI. Reference 14 is about breast (not breast cancer-related) LE. Is Figure 3f supposed to say "TPD index/little finger" instead of "TDP?" Figure 3's legend should say "Data points in b), c).....represent measurements in individual patients."

An approximate price for HSI should be named, and perhaps compared to an approximate price for US.

In the paragraph describing Methods for Patient-reported Outcomes, please change "It is a descriptive tool..." to "These questionnaires [plural] are descriptive tools...."

Please move description of TWI, TLE, and StO₂ (and spell them out) up to the beginning of the "Analysis of Hyperspectral images" section.

The hierarchical clustering, using $k=3$, for grading delta-LWR, might or might not work well for different study subjects/patients/secondary LE types, and may not be easy for device users to determine. This is shown in Figure 4 h and i-- for LE stage 2, particularly, users might or might not be able to translate HSI stage to LE stage. The authors are correct to state that "The proof-of-principle study does not yet define clear threshold values for delta-LWR that could be used in clinical practice to differentiate between stages of LE."

It appears that HSI can tell that LE is present, but may not be able to detect early (stage 0 or 1) LE. If it is possible to explain Figure 4d TWI + StO₂ indices images (what does red mean/indicate? What does green mean/indicate?), readers would benefit.

The Discussion should mention whether study subject/patient hydration level could affect results. It is difficult for the reader to discern whether HSI is simply a reflection of tissue water content (or oxygenation level), and how these parameters would align with what we know of LE-affected tissue biology (increased interstitial fluid at early LE stages, then increased adipose tissue in later stages, and increased fibrotic skin at later LE stages). It might be possible to speculate on any such correlations, or discuss how these parameters could cause the delta-LWR/stage 2 LE dispersed data.

Version 1:

Reviewer comments:

Reviewer #1

(Remarks to the Author)

All concerns have been addressed. No further comments.

Reviewer #2

(Remarks to the Author)

This interesting study describes the use of hyperspectral imaging for diagnosing breast cancer-related lymphedema (BCRL). The manuscript is well written. There are several points that need addressing:

- 1) short discussion of whether HSI can detect BCRL early, and if so, how early?
- 2) the first paragraph in the introduction has the word "sensibility" which should be "sensitivity," perhaps?
- 3) Consider adding palm view, as sometimes, palm alone can show dermal backflow when all other sites do not,
- 4) The use of HSI seems to require a lot of data processing that could be tough for average users to master,
- 5) what exactly is the "background?"--clothing, a black background?
- 6) the use of prescribed anatomical areas for surveillance could miss patches of affected tissue between the standard three observation sites (many BCRL arms do not appear to be uniformly affected along the arm's length)
- 7) "This suggests that some patients diagnosed with stage 2 LE did not exhibit physiological tissue changes measurable with the HSI method used in this study." discuss--due to stage 2 LE patients using compression garments with varying compliance? or adipose tissue retaining water?
- 8) "Depending on light source and spectral detector (define which types?), HSI enables a high-resolution, two-dimensional display...."
- 9) HSI's correlation with disease severity is 0.63, which is better, albeit, not by much, compared to BIS.
- 10) Reference 12 is indeed about breast LE (not breast cancer-related LE). While the inclusion criteria state that the patients are breast cancer patients, the focus of the reference (Dr. Brunelle is a leader in this field) is breast LE, where the breast itself, not the affected-side arm, swells. Many therapists must deal with this terrible form of LE. Unfortunately, breast LE is not well reported in the literature. Unless HSI is used to image breasts with breast LE, and there is some relevance to your work, you should remove/replace that reference.

Version 2:

Reviewer comments:

Reviewer #2

(Remarks to the Author)

Authors have addressed all points

Tübingen, 02.06.2025

Response to referees

Reviewer #1:

The authors investigated the potential of hyperspectral imaging (HSI) as a diagnostic tool for secondary lymphedema in breast cancer patients using the TIVITA HSI system. Their findings suggest that the lipid-to-water ratio (LWR) may serve as a promising objective marker for detecting lymphedema. Here are some comments:

1) Acronyms such as HSI, TWI, TLI, and LWR are defined multiple times throughout the manuscript. It would improve clarity and flow to define each acronym only at its first occurrence and use the acronym consistently thereafter.

Response: We removed the redundant definitions of the acronyms

2) While the study utilized a commercial HSI system and analysis software, additional detail on how TWI and TLI are calculated would enhance transparency and reproducibility. A brief description of the algorithm or underlying principles would be beneficial.

Response: Since the exact formulas for computing TWI and TLI are proprietary to the manufacturer of the HSI system we could not disclose them exactly. However, we've indicated a reference that is authored by the founders of Diaspective Vision, the manufacturer of the TIVITA Tissue System, where the algorithms for the indices and the reasoning behind them are explained in detailed.

We've added the following paragraph to the Methods section that explicitly states these points and briefly tries to describe what are the TIVITA indices we used and on which wavelength bands are they defined:

In the current study, TWI and TLI were used to test correlation with LE stage. StO₂ was also used together with TWI to distinguish body parts from the surrounding background in HSI images. The indices were computed from the spectral signal within individual wavelength bands and scaled to yield values between 0 and 100. The exact formulas and scaling factors are proprietary to the manufacturer of the HSI system, however many details about the algorithms can be found in Holmer et al. 2018⁸. In summary, TWI is a scaled ratio between the average signal in the 875-895 nm and 950-975 nm bands, TLI is a scaled mean of the second derivative of the signal in the 925-935 nm band and StO₂

is a scaled ratio between the second derivative of the signal in the 575-590 nm and 740-780 nm bands.

The limitations in sharing the exact formulas for TWI and TLI drove us to define the LWR measure, which captures similar information to TWI and TLI, but has a simpler formula that can be shared.

3) The manuscript states that "raw spectra were normalized to their overall intensity." Please clarify the normalization process. Specifically, does it involve referencing to white and dark calibration measurements, or is another normalization method used?

Response: We've reformulated this sentence to make more clear which normalization we're using and explicitly stated in the Methods section that no additional calibration measurements were used.

Spectra corresponding to body pixels in this area were normalized to their summed intensity using L1 normalization and averaged to obtain a representative spectrum for each measurement. Except the standard calibration with the color checking palette, performed at the beginning of each measurement day, no additional calibration measurements were considered for the normalization of spectra.

4) If the LWR is derived from spectral information around 920 and 970 nm, could similar results be achieved using just two narrowband measurements at these wavelengths? If so, this raises the possibility of implementing a more cost-effective and portable device rather than requiring a full HSI system.

Response: We've incorporated this suggestion in the Discussions.

Regarding hardware optimization, the results of this study suggest that only a few wavelengths are sufficient for tissue characterization in this particular application. Therefore, one could envision developing more cost-effective systems that measure light only at the wavelengths of interest. This would provide clinicians in less resource-rich regions, who cannot afford a full HSI system, with access to the same enhanced diagnostic capabilities.

5) Given that hyperspectral image data were collected from 58 patients, it seems feasible to perform statistical analysis of the measured spectral data or derived index (LWR).

Response: We've added to Figure 4i the results of the statistical comparison of delta-LWR values between adjacent LE stages and details about the tests in the Methods section.

6) The descriptions of the hyperspectral imaging (HSI) procedures and results appear to be somewhat redundant between the Methods and Results sections. It is recommended to streamline these sections by removing repetitive content, ensuring that methodological details are clearly described in the Methods, while the Results section focuses on presenting key findings and interpretations.

Response: We've removed paragraphs from the Results section that described (redundantly) methodological aspects and added more information in the Methods section.

Reviewer #2 (Remarks to the Author):

This manuscript reports a prospective, multi-center study, conducted by a very qualified group, to assess hyperspectral imaging (HSI) as a method for detection and classification of secondary LE. The figures are very neatly prepared, and the writing is good. The main finding of the study is that HSI images in 58 breast cancer-related study subjects showed Pearson's coefficients of 0.64 (compared to $r=0.57$ for skin thickness measured by ultrasound) correlating to LE stages.

Please check and perhaps replace some of the references, and do not rely on AI. For example, reference 2 is given to describe gynecological cancer-related LE, yet that paper reports on breast cancer-related LE, which is not considered gynecological.

Response: First, we would like to assure that we have not relied on AI in the selection of references. In the context of medical diagnostics, HSI is not yet a widely used diagnostic procedure. Therefore, it is not possible to find references on the use of HSI in lymphedema that relate exclusively to the background of breast cancer. Furthermore, depending on the country, the diagnosis, treatment, and follow-up of breast cancer and possible complications such as lymphedema fall under the responsibility of "gynecology". This is also the case in Germany, the country where the study was conducted. To nevertheless comply with the request of reviewer #2, we have decided to delete the sentence "*For certain gyneco-oncological interventions, a secondary LE incidence of up to 60% is reported, whereas generally the incidence is about 20% considering all gynecological tumors*" without replacement. This also deletes the corresponding references (formerly 1-3) that refer to lymphedema that relate to general gynecological tumors.

For reference 6, please use the ISL position consensus, not an anatomy book.

Response: We agree with reviewer #2 and changed the reference by the 2020 Consensus Document of the International Society of Lymphology "The diagnosis and treatment of peripheral lymphedema" (PMID: 32521126)

Reference 7 does not discuss HSI.

Response: The purpose of Reference 7 is not to discuss HSI (the HSI technology will be introduced later in the introduction), but rather to present an alternative current method of assessing LE severity (like the ISL classification, difficult to objectify). We would therefore like to retain this reference.

Reference 14 is about breast (not breast cancer-related) LE.

Response: We disagree in this point. Reference 14 (Breast lymphedema following breast-conserving treatment for breast cancer: current status and future directions. *Breast Cancer Res. Treat.* 2024, 204, 193–222) is indeed about breast cancer-related LE. The inclusion criteria for the study by Brunelle et al. clearly state: "patients who underwent breast conserving treatment (surgery ± radiation) for breast cancer". We would therefore like to retain this reference.

Is Figure 3f supposed to say "TPD index/little finger" instead of "TDP?"

Response: We've corrected the typo in Figure 3f.

Figure 3's legend should say "Data points in b), c).....represent measurements in individual patients."

Response: We've rephrased the sentence as suggested.

An approximate price for HSI should be named, and perhaps compared to an approximate price for US.

Response: We've added an upper limit price range for the HSI system we used in the Methods section.

Hyperspectral images were acquired using the TIVITA® Tissue System (Diaspective Vision, Germany), a medically approved HSI system with a cost below \$10.000.

In the paragraph describing Methods for Patient-reported Outcomes, please

change "It is a descriptive tool..." to "These questionnaires [plural] are descriptive tools...."

Response: We've changed the sentence as suggested.

Please move description of TWI, TLE, and StO₂ (and spell them out) up to the beginning of the "Analysis of Hyperspectral images" section.

Response: We moved the description of the TIVITA indices to the beginning of the 'Analysis of hyperspectral images' section in the Methods. Following comment 2) of reviewer #1, we provided more detail on how the indices are computed within the limits of what we can openly disclose, given that the exact formulas of the indices are proprietary to the HSI system manufacturer.

The hierarchical clustering, using $k=3$, for grading delta-LWR, might or might not work well for different study subjects/patients/secondary LE types, and may not be easy for device users to determine. This is shown in Figure 4 h and i--for LE stage 2, particularly, users might or might not be able to translate HSI stage to LE stage. The authors are correct to state that "The proof-of-principle study does not yet define clear threshold values for delta-LWR that could be used in clinical practice to differentiate between stages of LE." It appears that HSI can tell that LE is present, but may not be able to detect early (stage 0 or 1) LE.

Response: The reviewer #2 correctly pointed out that the clustering of ΔLWR may yield different thresholds with different patient cohorts. This is why we explicitly state this measure is not yet suitable for clinical applications. Our aim in defining the HSI-stage was to illustrate what a potential HSI-derived metric could look like. Such a metric would consider only objectively measured HSI information and could generate categories based on it. In the case of the HSI-stage we see that while correlated, the overlap with LE stage is not perfect. As both our manuscript and the reviewer point out, stage 1 LE is not detected with HSI and stage 2 LE patients exhibit a wide range of ΔLWR values, indicating that tissue physiology is heterogeneous among these patients. This result suggests that stage 2 LE comprises further subgroups of patient pathologies. In this context, HSI could be useful in distinguishing early from late stage 2 LE and guiding therapy accordingly. We added this point into the discussion part.

Patients with stage 0 and stage 1 lymphedema consistently presented low ΔLWR values, while those with stage 2 and stage 3 exhibited a broader range of values, reflecting increased tissue heterogeneity with advancing disease and the potential existence of pathological subcategories in the later stages of the disease.

If it is possible to explain Figure 4d TWI + StO₂ indices images (what does red mean/indicate? What does green mean/indicate?), readers would benefit.

Response: We extended the description of the *body-from-background* separation algorithm in the Methods section. In this context we wrote the following about the step involving TWI + StO₂:

For the background separation algorithm, a combination of TWI and StO₂ was selected, as these two tissue properties, water content and oxygenation level, proved most effective at separating imaged body parts from the surrounding background. First, the TWI and StO₂ indices were computed for each pixel in the HSI image to produce corresponding TWI and StO₂ images. These were then filtered using a Gaussian kernel of size (5,5) pixels to reduce noise. The index images were then normalized to their maximum value and summed up. The normalized summed values range from 0 to 2, with 0 representing low TWI and StO₂ indices for that pixel and 2 representing high indices.

In addition, in the histogram from Figure 4d we marked the peak of low TWI + StO₂ values as 'background pixels' and the peak corresponding to high values as 'body pixels', and the following was added in the legend of Figure 4d:

d) Steps of the algorithm for separating the patient's body from the background. From left to right: an image of the arm section, the sum of normalized TWI and StO₂ indices with blue/green hues corresponding to small values and orange/red hues corresponding to high values, the histogram of normalized summed indices showing a bimodal distribution that corresponds to background and body pixels, the mask of the arm section after applying Otsu thresholding to the histogram, and the masked image of the arm section. Rows represent individual examples for the hand, forearm, and upper arm.

The Discussion should mention whether study subject/patient hydration level could affect results.

Response: We've added this point in the Discussions as another limitation of the study.

The study did not consider the influence of patients' water intake behavior. Measurements were carried out before noon to minimize the influence of tissue swelling throughout the day. However, this does not eliminate the potential impact of individual hydration levels on the results.

It is difficult for the reader to discern whether HSI is simply a reflection of tissue water content (or oxygenation level), and how these parameters would align with what we know of LE-affected tissue biology (increased interstitial fluid at early LE stages, then increased adipose tissue in later stages, and

increased fibrotic skin at later LE stages). It might be possible to speculate on any such correlations, or discuss how these parameters could cause the delta-LWR/stage 2 LE dispersed data.

Response: Hyperspectral imaging is indeed an indirect reflection of the relative composition of lipids and water in superficial tissues. As correctly pointed out, tissue biology in lymphedema evolves from increased interstitial water content in early stages to progressive adipose and fibrotic changes in later stages. HSI is a summation parameter of a broad spectral wavelength range. By focusing on specific wavelengths with the indices used, we avoid unnecessary blurring and focus on water and lipids as potential influencing parameters. There is relevant evidence in the literature that beside lipid accumulation water content can also increase in advanced LE stages. At the cellular level, hypertrophic adipocytes (under inflammatory and metabolic influences) store increased water. Chronic inflammation is common in lymphedema tissue. This also leads to increased vascular permeability and lymphatic vessel formation, with increased interstitial water, even in adipose tissue. Particularly in stage II, these effects could cause a high degree of scatter and the delta-LWR/stage 2 LE dispersed data.

Additionally, we've added this point in the Discussion:

While early-stage LE is dominated by interstitial fluid, later stages involve adipose and fibrotic changes. Notably, adipose tissue in chronic LE may also retain water due to inflammation-induced permeability (Karastergiou, K., Smith, S. R., Greenberg, A. S., & Fried, S. K. (2012). Sex differences in human adipose tissues – the biology of pear shape. Obesity Reviews, 13(11), 877–886), potentially explaining the predominantly dispersed Δ LWR values in stage 2.

Response to reviewer comments (round 2):

Reviewer #2 (Remarks to the Author):

This interesting study describes the use of hyperspectral imaging for diagnosing breast cancer-related lymphedema (BCRL). The manuscript is well written. There are several points that need addressing:

1) short discussion of whether HSI can detect BCRL early, and if so, how early?

Response: As mentioned in the Results and in the Discussions, the HSI-based metrics were unable to distinguish between stage 0 and stage 1 LE. Therefore, this approach is better suited for advanced LE stages. We added an explicit statement about this in the discussions: '*..., suggesting that this metric could not detect early stage LE.*'

2) the first paragraph in the introduction has the word "sensibility" which should be "sensitivity," perhaps?

Response: We've corrected the word as suggested.

3) Consider adding palm view, as sometimes, palm alone can show dermal backflow when all other sites do not,

Response: Unfortunately, we're not able to measure the palm view on this set of patients any longer as the study has concluded. However, we're going to consider this suggestion and integrate this measurement in a follow-up study.

4) The use of HSI seems to require a lot of data processing that could be tough for average users to master,

Response: The data analysis steps in this proof-of-principle study aimed to identify HSI correlates of LE stages and potential threshold values to assist with diagnosis. In a possible medical device, these steps would of course be automated and integrated into an overall system.

A clinically adapted workflow could look like this:

1. Imaging the affected and control sides with HSI. The lipid-to-water ratio we introduced would automatically be computed for every hyperpixel in the image.
2. Determine a reference tissue patch on the control side. The user could either (a) manually select a reference tissue patch on the control arm via software or (b) rely on an algorithm to automatically make this selection. For example, the algorithm could isolate the body part from the background and compute an average LWR on the tissue in the

central part of the image. A combination of automatic reference selection and manual user refinement, if necessary, could also be a solution.

3. Visualize the relative LWR (Δ LWR) on the affected side with respect to the LWR of the reference patch from step 2. The user would then have the spatial distribution of the Δ LWR on the affected side. In conjunction with predetermined threshold values from the present study or from future studies, one could visualize the regions of the affected arm that exhibit structural tissue changes in comparison to the control arm.

This workflow could be integrated into the HSI imaging system software to provide users with direct feedback. Alternatively, it could be implemented as offline analysis software outside of the HSI system. The advantage of the latter solution is that it can be used with any HSI system capable of exporting measurements to an external medium.

5) what exactly is the "background?"--clothing, a black background?

Response: During the HSI measurements, the patients rested their arms on an examination table. The table's upholstery was covered in a dark-colored cloth. The algorithm used to separate body parts from the background was successful because it relied on water content and oxygenation levels, properties characteristic of living tissue, unlike upholstery material.

6) the use of prescribed anatomical areas for surveillance could miss patches of affected tissue between the standard three observation sites (many BCRL arms do not appear to be uniformly affected along the arm's length)

Response: We can confirm that also in our data the arm sections are in some cases not uniformly affected by LE. However, we proceeded with using the predefined anatomical position because one of our goals was to compare HSI with other techniques (e.g. circumference, ultrasound) for assessing tissue changes, which were applied to predefined anatomical positions. Selecting a tissue patch in a different location on the affected arm where LWR is highest would bias this comparison. As a result of this choice, our analysis may underestimate in some cases changes in LWR.

To investigate the influence of the choice of standardized anatomical areas on the outcome of our analysis, we repeated the analysis pipeline with some modifications. For the LE-affected side, rather than selecting the tissue patch at the centre of the image, we chose a patch of the same size in a position that yielded the highest LWR value. For the control side, we maintained the central position of the tissue patch. As shown in the figure below, this led to an overall increase in the Δ LWR values, as all points are located above the main diagonal.

Comparison of the Δ LWR values when the tissue patch is located at the center of the image for both affected and control sides (x-axis values) with the scenario where the tissue patch is located at a position that yields the highest LWR value on the affected side and the center of the image on the control side (y-axis values).

However, the increase in Δ LWR values did not lead to a considerable increased correlation between the HSI-derived metrics and the diagnosed LE stage. Correlation with LE stage (Fig. 4j) increased from 0.63 to 0.64 for Δ TWI, and from 0.64 to 0.65 for the HSI-stage. Given these minor increases, we argue that, for simplicity and consistency with the location of the other measurements performed in the study, it is best to keep the current strategy of selecting the center tissue patch.

7) "This suggests that some patients diagnosed with stage 2 LE did not exhibit physiological tissue changes measurable with the HSI method used in this study." discuss--due to stage 2 LE patients using compression garments with varying compliance? or adipose tissue retaining water?

Response: Patients with an indication to wear support stockings were asked to take them off the evening before and only put them on again after the measurement on the following day. We've added this information in the discussion. Additionally, we added the following sentence after the paragraph listing the variables that were not specifically controlled for in the study: '*It is conceivable that some of the heterogeneity in Δ LWR values for stage 2 LE patients could be attributable to these variables*'.

8) "Depending on light source and spectral detector (define which types?), HSI enables a high-resolution, two-dimensional display...."

Response: We meant to convey with this sentence that the resolution of hyperspectral data depends on the resolution of the spectral detector and the wavelength bandwidth of the light used to illuminate the tissue. We rephrased the sentence to simply state what the output of HSI is, namely '*a high-resolution, two-dimensional representation of a surface, incorporating*

information from underlying layers'. The important parameters of the HSI system are listed in the Methods section.

9) HSI's correlation with disease severity is 0.63, which is better, albeit, not by much, compared to BIS.

Response: One potential advantage of HSI over BIS is that HSI offers a spatial distribution of tissue properties. Extra analysis carried out in response to reviewer's point 6 showed that considering the tissue patch with the highest LWR on the affected side did not significantly increase the correlation between HSI-derived metrics and LE stage. This suggests that, in our patient cohort, the spatial distribution of LE-induced tissue changes did not influence the results. However, we cannot rule out the possibility that the spatial distribution of tissue changes may be important for future investigations, or applications.

10) Reference 12 is indeed about breast LE (not breast cancer-related LE). While the inclusion criteria state that the patients are breast cancer patients, the focus of the reference (Dr. Brunelle is a leader in this field) is breast LE, where the breast itself, not the affected-side arm, swells. Many therapists must deal with this terrible form of LE. Unfortunately, breast LE is not well reported in the literature. Unless HSI is used to image breasts with breast LE, and there is some relevance to your work, you should remove/replace that reference.

Response: Following the recommendation of the reviewer, we removed the sentence "Limb circumference measurements, for example, rely heavily on operator experience and lack sensitivity to early or subclinical stages of the disease" and the corresponding reference from the manuscript.